# Hybrid simulation modelling of networks of heterogeneous care homes and the inter-facility spread of Covid-19 by sharing staff

**Le Khanh Ngan Nguyen**\*, **Itamar Megiddo**, **Susan Howick**

Department of Management Science, Strathclyde Business School, University of Strathclyde, Glasgow, United Kingdom

\* nguyen-le-khanh-ngan@strath.ac.uk

## Abstract

Although system dynamics [SD] and agent-based modelling [ABM] have individually served as effective tools to understand the Covid-19 dynamics, combining these methods in a hybrid simulation model can help address Covid-19 questions and study systems and settings that are difficult to study with a single approach. To examine the spread and outbreak of Covid-19 across multiple care homes via bank/agency staff and evaluate the effectiveness of interventions targeting this group, we develop an integrated hybrid simulation model combining the advantages of SD and ABM. We also demonstrate how we use several approaches adapted from both SD and ABM practices to build confidence in this model in response to the lack of systematic approaches to validate hybrid models. Our modelling results show that the risk of infection for residents in care homes using bank/agency staff was significantly higher than those not using bank/agency staff (Relative risk [RR] 2.65, 95% CI 2.57–2.72). Bank/agency staff working across several care homes had a higher risk of infection compared with permanent staff working in a single care home (RR 1.55, 95%CI 1.52–1.58). The RR of infection for residents is negatively correlated to bank/agency staff's adherence to weekly PCR testing. Within a network of heterogeneous care homes, using bank/agency staff had the most impact on care homes with lower intra-facility transmission risks, higher staff-to-resident ratio, and smaller size. Forming bubbles of care homes had no or limited impact on the spread of Covid-19. This modelling study has implications for policy makers considering developing effective interventions targeting staff working across care homes during the ongoing and future pandemics.

## Author summary

We developed an integrated hybrid simulation model to evaluate the impact of bank/ agency staff on the spread of Covid-19 across a network of heterogeneous care homes and the effectiveness of interventions targeting these staff. The study methods advance infectious disease hybrid simulation modelling, and in particular, we demonstrate approaches for validation and confidence building of hybrid models; an under-researched area. Our

**Data Availability Statement:** Data on cases were obtained from the GOV.UK Coronavirus (COVID-19) in the UK (https://coronavirus.data.gov.uk/

details/cases). Other data are available at https://github.com/lenguyen3150/Bank_staff_COVID-19.

**Funding:** LN and IM were supported by the National Institute for Health Research (Policy Research Programme, Responsive Operational Research Facility, PR-R17-0916-21002). The funders had no role in study design, data collection and analysis, decision to publish, or preparation of the manuscript. The views expressed in this publication are those of the author(s) and not necessarily those of the NHS, the National Institute for Health Research or the Department of Health and Social Care.

**Competing interests:** The authors have declared that no competing interests exist.

model findings fill a research gap with policy implications for care homes, which are heavily reliant on bank/agency staff due to shortages of staff. Our findings align well with existing observational study evidence, including that that using bank/agency staff increases the risk of infection for residents and that bank/agency staff have a greater risk of infection compared with permanent staff working in single care homes. Using bank/agency staff has the greatest impact on infections in care homes with lower intra-facility transmission risks, higher staff-to-resident ratio, and smaller size. Testing bank/agency staff is particularly important while forming smaller bubbles of care homes and restricting staff to only work within a bubble has limited impact on the spread of Covid-19. The model is being used to assist decision-makers from the Department of Health and Social Care in the UK.

## Introduction

According to recent evidence, staff working across different care homes are at a greater risk of infection than those working in a single care home, and their use significantly increases the risk of outbreaks among residents [1,2]. Studies in English care homes showed that staff working across different care homes had a three-fold (95% confidence interval [CI], 1.9–4.8) higher risk of infection than those working in single care homes. Further, frequent employment of agency staff increased the odds of infection for residents by 1.65 (95%CI, 1.56–1.74) [1,2]. New legislation may ban staff from working in more than one care home in an attempt to halt the spread of Covid-19 [3]. These types of interventions need to be thought through as they may lead to unintended consequences such as difficulties in recruiting staff. They also need to be balanced against outcomes that are not related to Covid-19.

Working across different care homes and other healthcare facilities has been a common practice among care home staff for reasons of flexibility, work-life balance, and extra income. A recent survey in the US showed that 17% of long-term care workers had a second job and 60% held double- or triple-duty caregiving roles [4]. Furthermore, care homes in many countries, including the UK and the US, are heavily dependent on the use of temporary bank or agency staff due to the long-standing problem of staff shortages in the health and social care sector, which has been worse amid the pandemic [5–7].

Knowledge is limited on the extent to which staff work in multiple care homes and contribute to spreading infection as well as which interventions effectively target this group. While interventions that limit staff movement across care homes may reduce infection, they can also reduce the number of staff available in a given care home. Understaffing care homes could lead to lower quality of care for residents, and a lower staff-to-patient ratio can also increase transmission within care homes as each staff member needs to interact with more residents [8,9]. Care home workers hired from agencies typically have zero-hour contracts and a low income, and thus, they have little power to influence policy proposals to ban their movement. Such a proposal could lead them to leave their positions or the sector due to financial instability and job insecurity, threatening the closure of care homes [10].

Modellers have used both top-down (system dynamics [SD]) and bottom-up (agent-based modelling [ABM]) approaches to study SARS-CoV-2 dynamics and interventions at the country level and in specific settings, which have contributed to Covid-19 policy decisions. SD is suitable and preferable to provide a holistic view of systems to support policymakers in making strategic decisions that influence a large population. Examples include SD models assessing the effects of mass social isolation policies, lifting restrictions, mass vaccination and

prioritization, and the control of outbreaks in multiple care homes [11–16]. Modellers also find SD useful to quickly evaluate infection prevention and control [IPC] measures implemented in care homes and the relative contributions of all importation routes without needing data on the interactions and processes taking place at the micro-level within each facility [17,18]. By contrast, ABM is well-suited for capturing the detail and microstructure of intricate settings such as a care home, including the complexity, heterogeneity, and stochasticity of interactions between different individuals (residents, staff, and visitors) within a care home. ABM also enables interventions such as cohorts and targeted testing to be explicitly modelled. A number of studies have adopted ABM to model SAR-CoV-2 transmission and interventions within a single care home [19–24].

Each simulation approach has different advantages and disadvantages, and by combining different simulation modelling methods in a hybrid simulation model we can address Covid-19 questions and study systems and settings that are difficult to study with a single approach. Additionally, as each method considers a problem from a different perspective, intricate and interconnected pandemic related problems can benefit from the complimentary view and deeper insight gained from using multiple simulation methods together [7]. However, despite the growing interest in hybrid simulation and its potential benefits, a number of challenges have been raised, such as the lack of a clear methodological approach to develop hybrid simulation models, the requirement of multiple expertise, and the increasing cost of using more than one software package [25]. As a result, the use of hybrid simulation is still limited and only a small number of hybrid simulation models have studied the dynamics of infectious disease transmission across different settings and in fragmented populations [26–30].

In this study, we develop a hybrid SD-ABM simulation model to investigate the impacts of staff working across different care homes as well as interventions to mitigate these impacts. Specifically, we consider the impact of reducing or stopping the use of bank/agency staff, weekly PCR testing, and creating bubbles of care homes on the spread of Covid-19. Care home bubbles restrict bank/agency staff to work only within a specific group of care homes that are designated as one bubble. Our model provides a tool for exploring the interaction between interventions as they can undermine or enhance each other when implemented simultaneously. It also helps study the variations in the impact of using bank/agency staff on individual care homes in different network compositions. We adapt methodology from SD and ABM practice and theory to build confidence for validating our hybrid model.

Our model is being used to assist policy-makers from the UK Department of Health and Social Care who have been considering effective interventions targeting bank/agency staff. We use data from care homes in Lanarkshire as a case study for parametrization.

## Materials and methods

### Ethics statement

This research was approved by the Ethics Committee of the Department of Management Science, Strathclyde University.

### Model structure

We developed an integrated hybrid simulation model that combined SD and ABM. This section provides an overview of the model structure, and the ODD (Overview, Design Concepts, and Details) protocol (S1 Appendix) provides further detail. The hybrid model contained three modules built using either SD or ABM: Network (ABM), Temporary Staff (ABM), and Intra-facility (SD) (Fig 1).

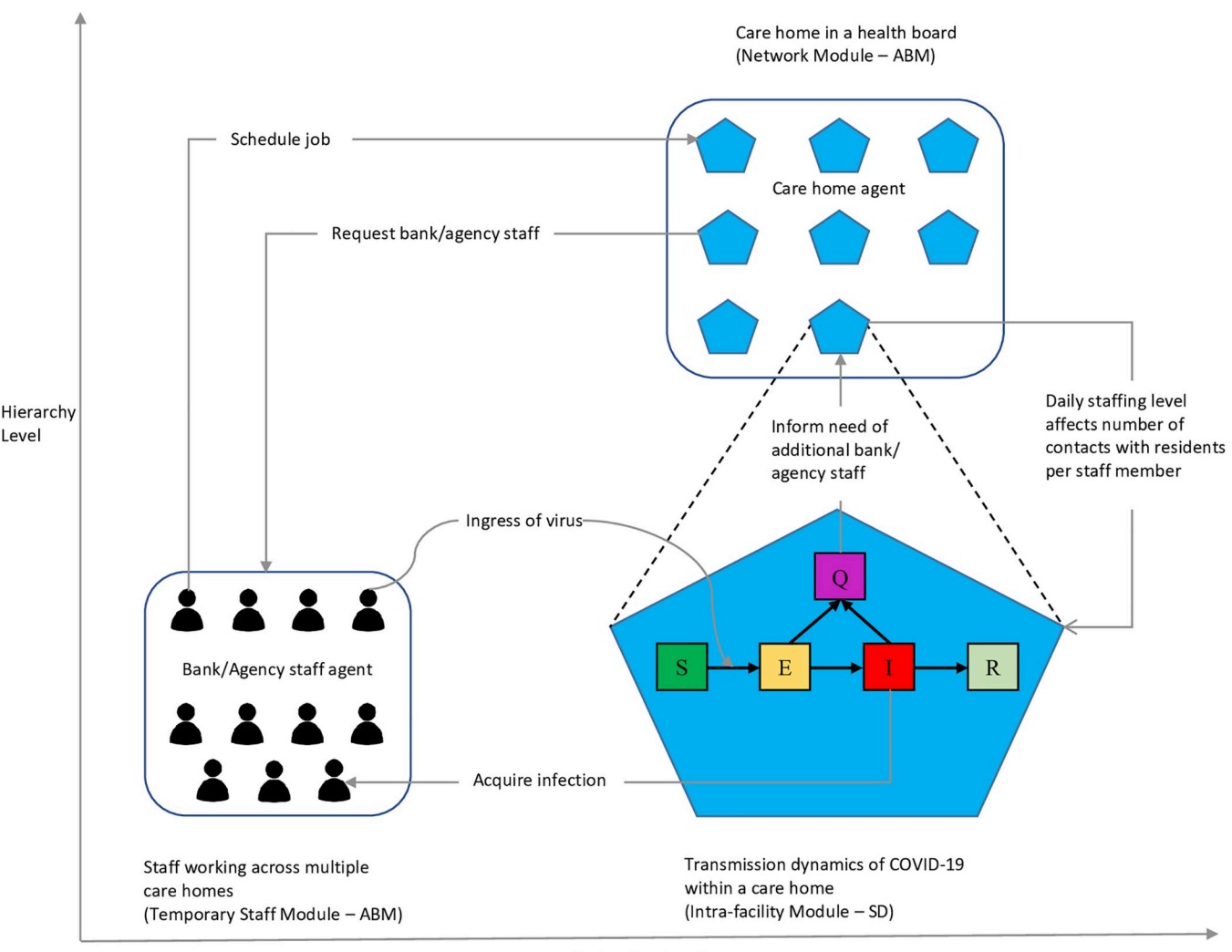

**Fig 1. Architectural design of the integrated hybrid SD-AB model comprising three modules**

**Network module.** The Network module models the constituent care homes in a network that share bank/agency staff. In this module, care home agents are characterised by their resident population size, staffing level, bank/agency staff use, and intra-facility transmission rates (Table A in S1 Appendix). ABM is appropriate to build this module to address our questions as it can capture the heterogeneity in care homes' ingress risk and intra-facility transmission dynamics, which affect the homes' risk of experiencing outbreaks. Bank/agency staff shared between care homes can spread the virus from one facility with a current outbreak to other facilities with no cases of infection causing them to experience outbreaks. ABM is also more flexible than SD for reflecting any changes in network composition and enables the explicit modelling of interventions such as creating bubbles of care homes. The composition of a network and such interventions may affect the extent to which the virus spreads across constituent care homes.

**Temporary staff module.** In the Temporary Staff module, bank/agency staff members are modelled as agents whose state variables are described in Table A in S1 Appendix. As they are scheduled to work in different care homes on a daily basis following specific rules, it is

important to consider them at the individual level to capture the stochasticity of their movement across care homes. Chance events such as a number of care homes having outbreaks in low community infection prevalence may emerge from the collective movement actions of bank/agency staff agents. Furthermore, ABM offers more flexibility for explicitly incorporating the restriction of movement on bank/agency staff within a bubble of care homes.

**Intra-facility module.** A stochastic SD Intra-facility module is embedded within each care home agent of the Network module and represents the transmission dynamics of Covid-19 in each care home. In this module, individuals were aggregated based on their role (residents or staff members), state of infection, testing and isolation status (Fig C in S1 Appendix). Although we appreciate that the heterogeneity of individual traits and behaviours and the detailed operational structure are important in characterising the transmission dynamics within settings such as care homes, investigating the extent of their impact on the spread of the virus is not the purpose of this hybrid model and has been studied in our previous work [24]. To address our questions on transmission across a heterogenous network mediated by bank/agency staff, it is preferable to simplify the model by reducing the transmission dynamics complexity within each care home. The use of stochastic SD in this case also leads to lower computational intensity while still allowing for the stochastic transmission dynamics and the extinction of the virus and, thus, capturing the risk of outbreaks in each care home. Furthermore, as the investigated problem focuses on the transmission via bank/agency staff across care homes in a network, each care home is viewed as a sub-system from a holistic perspective. Each care home's macro characteristics and behaviours, rather than individuals' characteristics and behaviours, are of importance for decision makers who manage a network of care homes at a regional level. We described the equations for stocks and flows of this module in Section 7.4 of S1 Appendix.

**Temporal scale.** The agent-based modules Network and Temporary Staff run at a daily time step as epidemiological data are collected on a daily basis and this is also the unit of time commonly used to describe clinical characteristics of Covid-19 in the literature. The stochastic SD module Intra-facility is theoretically based on continuous time in which the time step $dt$ represents an infinitesimal time-step [31]. In practice, the module runs at a small finite time step $dt$ of $1/2^7$ days for good numerical results for a continuous model. The modules exchange information daily to capture transmission dynamics across care homes. As bank/agency staff are rostered daily, it is important to update their infection state and the state of SD modules in affected care homes on this time scale. Simulations are 90-day time steps long as this covers the period for planning response strategies to contain the spread of Covid-19. However, we also ran the model for 180 days to assess the robustness of the findings for a longer time period.

## Updating rules

We refer to bank/agency staff in this paper as non-permanent staff under bank, agency, temporary, casual, and non-guaranteed hours contracts. Demand in care home $i$ for bank/agency staff on a given day during the pandemic is described by $N_{B,i}$ as follows:

$$N_{B,i} = N_{BN,i} + \frac{Q_{S,i}(N_{W,i} - N_{BN,i})}{N_{S,i}}$$

$$N_{BN,i} \sim Poisson(\alpha N_{W,i})$$

$N_{BN,i}$ *describes* the number of bank/agency staff required in normal circumstances prior to the Covid-19 pandemic due to ongoing staff shortage and absence of staff for reasons such as

holidays, unfilled vacancies, and sickness. The parameter $\alpha$ is the average percentage usage level of bank/agency staff in all care homes in the network. The value of $\alpha$ is between 5% and 20% across various areas in the UK [5,32,33]. In base case scenarios, we set $\alpha$ to 10%. $N_{W,i}$ denotes the desired number of staff members on duty per day when care home $i$ operates at full capacity. $N_{S,i}$ is the number of permanent staff members of care home $i$. The component $Q_{S,i}(N_{W,i}-N_{BN,i})/N_{S,i}$ represents the number of bank/agency staff agents required to cover for permanent staff members self-isolating due to Covid-19 ($Q_S$).

For each care home agent $i$, *chosen* randomly, bank/agency staff agents who have not been allocated to any other care home and are not self-isolating are allocated based on the following two rules one by one until the demand of this care home is fulfilled.

- Rule 1 with probability $\eta$: A randomly chosen bank/agency staff agent is allocated. In base case simulations, $\eta$ is set to 0.5 based on discussion with bank/agency staff members and care home managers in Lanarkshire.

- Rule 2 with probability $(1-\eta)$: The bank/agency staff agent with the largest value of *WorkRecord[i]* is allocated. The rule describes a desire by care homes to utilise the same bank/agency staff and by these staff to work in the same care home. *WorkRecord* is initiated by a warm-up period of 90 days without infections in each simulation run. It reaches a steady state after this period.

If the number of available bank/agency staff is insufficient to fill the required positions, care homes will be understaffed for that day.

The number of contacts with staff per resident remains unchanged based on the implicit assumption that the overall care home workload does not change and, therefore, is not affected by the daily staff-to-resident ratio. This means that staff on duty will have to carry out extra workload to maintain the quality care delivered to residents. This was modelled by updating the daily number of contacts with residents per staff member used in the SD Intra-facility module (Section 7.3 in S1 Appendix).

The daily number of infectious bank/agency staff members increase the forces of infection for susceptible residents and susceptible permanent staff in the Intra-facility module (Section 7.4 in S1 Appendix). At the end of the day (ABM time-step), susceptible bank/agency staff acquire infection via interactions with infectious residents and other staff members at a rate which is equal to the force of infection in staff in the care home where they have worked (Section 7.5 in S1 Appendix).

## Experiment scenarios

We considered the impact of different intervention scenarios on the spread of Covid-19 within a network of 12 care homes (network A), which consists of a total of 780 residents, 960 permanent staff members, and 107 (10%) bank/agency staff members. The proportions of bank/agency staff were varied by scenario, but the total number of staff remained the same across scenarios. Sizes and staff-to-resident ratios of constituent care homes were determined based on the empirical distributions of 84 care homes in Lanarkshire and the same characteristics were used for all simulations (Table B in S1 Appendix). Data in Lanarkshire also reflected the proportions of care homes by size ranges in the UK [2,34]. All residents and staff were susceptible at the beginning of simulations.

We assumed that care homes operate at their full capacity for the entire simulated period. The assumption helped avoid any distortion to the network composition in terms of constituent care homes' size and staff-to-resident ratio. Each care home in a network implemented the following interventions: hand hygiene and use of personal protective equipment, social

distancing, testing and isolation upon admission and re-admission of residents, closure to visitation, and weekly PCR testing of permanent staff (80% compliance) [35,36]. We assumed that bank/agency staff and permanent staff have the same risk of infection acquiring in the community. We examined the impact of different average usage levels of bank/agency staff with no pandemic ($\alpha$): 0%, 5%, 10%, 15%, and 20%. We considered interventions that create similarly sized care home bubbles of size two, three, four, or six care homes. Care homes were grouped into bubbles randomly in the base-case simulations under the assumption that this would be done based on care homes' geographic location in reality. We also explored a scenario in which care homes are grouped based on their resident population size and staff-to-resident ratio. Bank/agency staff agents were grouped into these bubbles so that the ratios of bank/agency staff to total staff were as equal as possible across the bubbles. We also explored the impact of these interventions given different compliance rates to weekly PCR testing among bank/agency staff ranging from 0% to 80% in 20% increments. Additionally, we examined the effect of different levels of staff shortage (0%, 5%, 10%, 15%, and 20%) and compared scenarios in which bank/agency staff were used to compensate for the shortage and ones in which they were not.

We assessed how sharing bank/agency staff affects individual care homes' Covid-19 outcomes and how care home characteristics affect these outcomes. We performed the experiments for three other hypothetical networks B, C, and D that comprise the same number of residents and staff members but have different compositions. Network B, which consists of homogeneous care homes in terms of size and staff-to-resident ratio, was used to examine the effect of using bank/agency staff on individual care homes with different intra-facility transmission risk drawn from a distribution. This heterogeneity represents different levels of adherence to care home interventions and other care home characteristics (e.g., architecture and operation) that we abstract from the SD Intra-facility module. Care homes in network C are homogeneous in size and heterogeneous in staff-to-resident ratio, and those in network D are heterogeneous in size and homogeneous in staff-to-resident ratio. We used networks C and D to disaggregate the impact of the heterogeneity in size and staff-to-resident ratio on the model results. Experiments with these networks also helped examine the robustness of model results to changes in network composition.

## Outcomes

We considered the cumulative number of infections in residents, permanent and bank/agency staff (medians and confidence intervals [CIs], interquartile ranges, and distributions) and the probabilities of outbreak occurrence (i.e. presence of at least two infected residents) in m care homes (m = 1, 2,. . ., 12). The model is stochastic and yields a distribution of possible outputs for each outcome for each set of input parameters, describing first-order uncertainty, and thus requires a large number of simulations to capture the system behaviour. We ran 1,000 simulations for each scenario since the median outputs of each outcome converged after this number of simulations. With this number of simulations, the 95% CIs of the median outputs for the cumulative number of infections was ± one infection per 1,000 people and the probability of outbreak occurrence was ± 5%.

## Statistical analysis

We used the Wilcoxon test at a significance level of $\alpha = 0.05$ to perform hypothesis testing for the difference between scenarios in the median cumulative numbers of infections in residents after 90 days. We also adopted the Bonferroni correction method in which the *p*-values were multiplied by the number of tests to counteract the potential for type 1 error in multiple

comparisons. Where there was no statistical significance in median outputs of this outcome between scenarios, we performed the Kolmogorov-Smirnov [KS] test to identify the difference in distributions of the outputs. We also calculated the relative risk [RR] of infection for residents and the RR of outbreaks for different pairs of scenarios and the RR of infection in bank/agency staff to permanent staff for each scenario.

## Confidence building (Verification and validation)

Our simulation model was built in Anylogic PLE 8.7.5, a multimethod simulation modelling tool that combines graphical modelling and Java code, and analysis was carried out in R version 1.4.1717. We gained confidence in the modules and the overall hybrid model using several approaches: code verification (S2 Appendix), white-box validation (including face validation, interface validation), black-box validation, and sensitivity and uncertainty analysis. We adapted these methods from both SD and ABM practices responding to the lack of systematic approaches for building confidence in hybrid simulation models [25].

In white-box validation, we developed individual modules and the hybrid model by triangulating insights from the literature, secondary data, and interviews and discussions with care home stakeholders, including representatives from Health and Social Care Partnership Lanarkshire, Public Health Lanarkshire, and staff and managers of care homes in Lanarkshire. The model was presented to and challenged by the Scottish Government Data Analysis Research Group. This helped ensure that the model structure and parameters sufficiently represented the investigated system and that our assumptions were appropriate for the model's purposes. We, in consultation with the stakeholders, continuously assessed the selection of SD and ABM for each module and the design of the hybrid model throughout the modelling process to ensure the appropriate level of abstraction for each part of the system. For instance, we compared the stochastic SD Intra-Facility module with parallel deterministic SD and ABM models providing complementary representations of the same system at a different level of abstraction (S2 Appendix). This approach helped gain plausible explanations of the system behaviour and understand any differences in outcomes resulting from the use of these different simulation modelling methods. Additionally, we assessed the design of the SD-ABM interfaces—in terms of what and how information is exchanged between the modules—and updating rules to ensure the synchronisation of the modules.

In black-box validation, we adopted the pattern-oriented modelling approach [37] to assess the model's ability to reproduce the following patterns observed in care homes in the UK: i) the higher risk of infection for residents and staff in care homes that frequently use bank/agency staff compared with ones that do not use them [2]; ii) the higher risk of infection for bank/agency staff compared with permanent staff in care homes that frequently use bank/agency staff [1,2]; iii) the higher risk of outbreaks in care homes that frequently use bank/agency staff compared with ones that do not use them [2,38,39]; iv) the risk of outbreak occurrence in care homes specified by their size and staff-to-resident ratio [38,40,41]. Patterns i, ii, and iii, which reflect the impact of agency/bank staff use upon the spread of Covid-19 across care homes within a network, are important to clarify that our model is useful for its purposes. Pattern i, ii, and iii help validate the behaviours of the overall system. Patterns iv addresses the validity of the sub-systems' behaviour (care homes) when accounting for their interactions via bank/agency staff. We identified the studies to which we compared our modelling results by a systematic search of PubMed, the WHO Covid-19 database, and medRxiv on June 25, 2021 (S2 Appendix).

We performed a global sensitivity analysis for parameter uncertainty in the base case scenario and uncertainty analyses for structural and characteristic changes of the model to

establish the robustness of the results and their uncertainty. The global sensitivity analysis parameter probability distributions are summarised in Table A in S4 Appendix. We ran 10 iterations for each of the 10,000 sets of samples generated using the Latin hypercube sampling (LHS) method (i.e. 100,000 simulations in total). The calculated partial rank correlation coefficient (PRCC) determined the strength of the relationship between each LHS parameter and each outcome measure. For model structural and characteristic uncertainty, we examined the impact of different network composition (networks A, B, C, and D) as described above and the heterogeneity of care homes' intra-facility transmission risk upon the model outcomes which we explain below. We also explored the model sensitivity to different network sizes by considering networks of 24 and 6 care homes.

We conducted each experiment in three different scenarios of intra-facility transmission risk. In the first scenario, care homes in a network have the same average per-contact transmission probability of 0.02 calibrated for a single care home in Lanarkshire. The studies of transmission risk in similar settings reported similar values [42]. The corresponding $R_o$ of 4.02 in a care home was in line with the base case $R_o$ of 4.04 used in a study of Covid-19 spread in a long-term care facility in France [22]. The intra-facility transmission risk is affected by institutional and operational factors such as physical layout, ventilation, provided care services, cohorting, and adherence to interventions, and therefore, likely to be heterogeneous among care homes. For example, care homes providing nursing care have been more likely to have infected residents, possibly owing to their residents' higher level of dependency requiring closer contact with care staff [38]. In the second scenario to reflect this heterogeneity, the average per-contact transmission probability in each care home was drawn from a Beta distribution (shape 1 = 5, shape 2 = 266). In the third scenario, we used Beta (shape 1 = 2, shape 2 = 117) to incorporate greater heterogeneity in the transmission risk across care homes. To obtain these distributions, we calibrated the hybrid model with heterogeneity in transmission risk to the time series of average daily infection prevalence produced by the baseline model with homogeneous transmission risk (Fig B in S2 Appendix). The objective function minimizes the sum of squared errors and uses Tabu search and scatter search, which make less use of randomization and greater use of strategic choices, and therefore, are unlikely to be trapped in a local optimal solution [43]. We varied the two Beta distribution parameters and held other parameters constant.

## Results

### Impact of different usage levels of bank/agency staff

The usage level of bank/agency staff had a significant impact on the risk of infection for residents and the risk of outbreaks across care homes (Fig 2; Table A in S3 Appendix). When bank/agency staff were not tested weekly, the RR of infection for residents in care homes using an average of 10% bank/agency staff compared with those in care homes not using bank/agency staff was 2.65 (95%CI 2.57–2.72). When we set the average level of bank/agency staff to 20% of total staff, this RR of infection almost doubled (5.17, 95%CI 5.03–5.30). The RRs of outbreaks in care homes using 10% and 20% bank/agency staff to those not using bank/agency staff were 3.76 and 5.64 respectively (95%CI 3.58–3.96 and 5.37–5.92).

The magnitude of the effect of using bank/agency staff significantly reduced when they were more compliant with the weekly PCR testing intervention. However, when bank/agency staff's compliance to weekly testing was as high as permanent staff's (80%), using bank/agency staff still increased the risk of infection for residents and the risk of outbreaks in care homes. The RRs of infection for residents in care homes using an average of 10% and 20% bank/agency staff compared with those in care homes not using bank/agency staff were 1.28 (95%CI

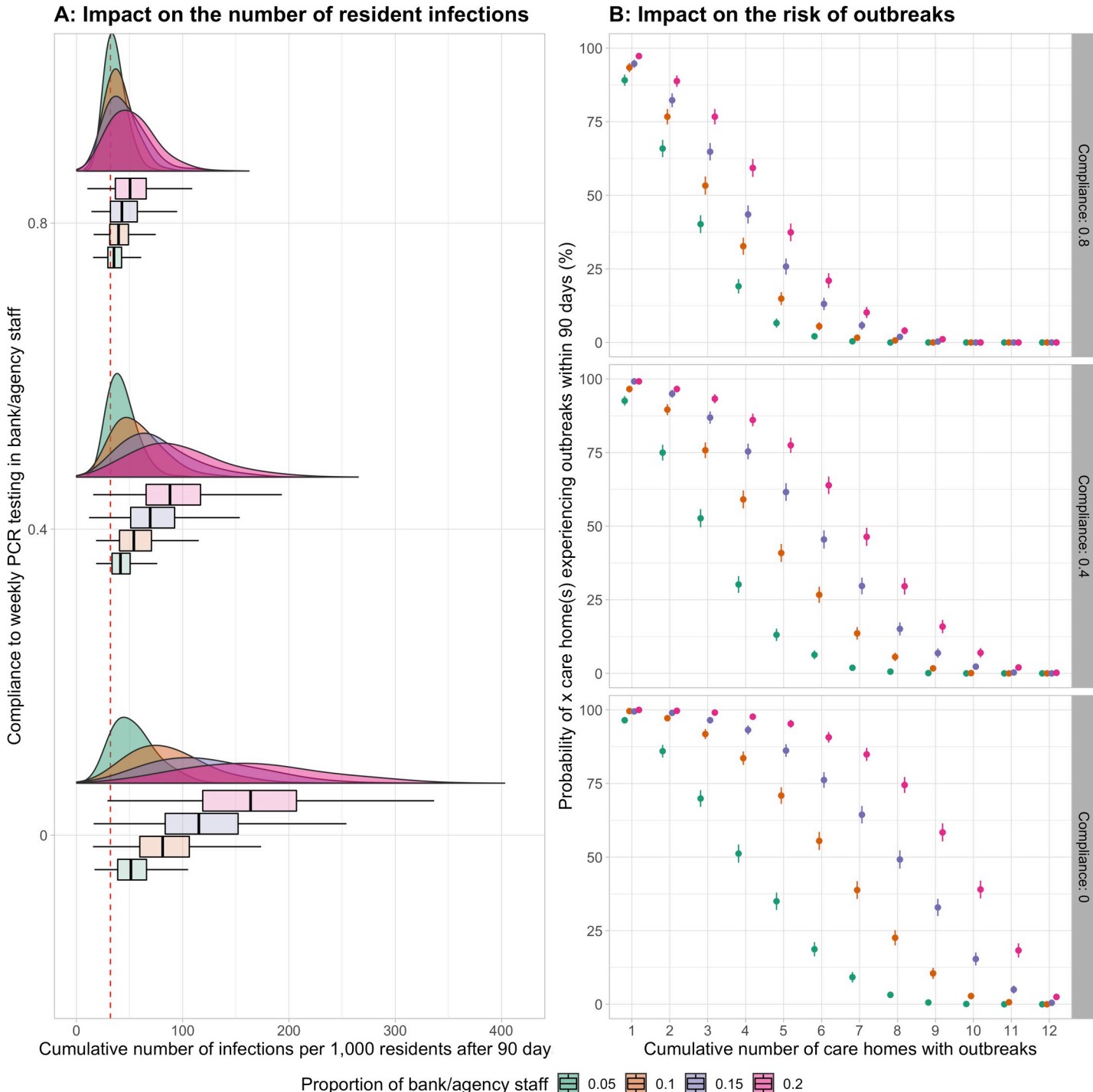

**Fig 2. Impact of using bank/agency staff with different compliance rates to weekly PCR testing.** A. On the cumulative number of infected residents after 90 days: Red dashed line denotes the median cumulative number of infected residents when care homes do not use bank/agency staff. Results are for 1,000 simulations in each scenario. Boxplot: middle–median; lower hinge– 25% quantile; upper hinge– 75% quantile; lower whisker = smallest observation greater than or equal to lower hinge— 1.5 * IQR; upper whisker = largest observation less than or equal to upper hinge + 1.5 * IQR. B. On the risk of outbreak occurrence across care homes within 90 days. The risk of outbreak occurrence (point) is the proportion of simulations where outbreaks occur in 1,000 simulations for each scenario. Line range denotes the 95% CI of this outcome.

1.25–1.31) and 1.64 (95%CI 1.60–1.68) respectively. The corresponding RRs of outbreaks were 1.83 (95%CI 1.73–1.94) and 2.48 (95%CI 2.35–2.61).

Bank/agency staff working across multiple care homes also had a higher risk of infection than permanent staff working in single care homes (Fig A in S3 Appendix). When the average usage level of bank/agency staff was 10% of total staff, the RRs of infection for bank/agency staff compared with permanent staff were 1.55 and 1.35 in the scenarios of 0% and 80% compliance to weekly testing respectively (95%CI 1.52–1.58 and 1.32–1.38). When the average usage of bank/agency staff increased to 20% of total staff, these RRs of infection were 1.98 and 1.48 in the scenarios of 0% and 80% compliance to testing respectively (95%CI 1.95–2.01 and 1.46–1.51).

## Impact of using bank/agency staff upon individual care Homes with different characteristics

The impact of using bank/agency staff on the risk of outbreaks varied across care homes with heterogeneous characteristics in a network. The use of bank/agency staff was more impactful in care homes with lower intra-facility transmission risk, higher staff-to-resident ratio, and/or smaller resident population size (Fig 3). The RR of outbreaks in care homes using bank/agency staff (at 10%) was negatively correlated to their intra-facility transmission risk (Fig 3A). The RR for the care home with the lowest transmission risk in the network was 17.3 (95%CI 9.47–31.5), whilst that for the care home with the highest transmission risk was 1.11 (95%CI 1.08–1.15). We observed a similar trend for the RR of care homes with different sizes (RR for 10 residents 19.3, 95%CI 4.12–61.6; RR for 160 residents 1.75, 95%CI 1.64–1.86) (Fig 3C). By contrast, the RR of outbreaks in care homes using bank/agency staff was positively correlated to their staff-to-resident ratio (for the ratio of 0.77: RR 2.46, 95%CI 2.16–2.8; for the ratio of 1.77: RR 11.0, 95%CI 8.26–14.5) (Fig 3B).

## Impact of forming bubbles of care homes

**Randomly grouping care homes into similarly sized bubbles.** When bank/agency staff were not tested weekly, creating smaller bubbles of care homes and restricting bank/agency staff from working across these bubbles slightly reduced the spread of Covid-19 across care homes. Forming bubbles of two to four care homes reduced the cumulative number of infections by six (95%CI 5–7) per 1,000 residents after 90 days. When the weekly PCR testing of bank/agency staff was implemented, creating bubbles of care homes had no statistically significant effect on the cumulative number of infected residents (pairwise Wilcoxon tests: p>0.1, KS tests: p>0.05 except for the pair of bubble size of two/three and 12 in the testing compliance of 20%: p<0.001).

**Grouping care homes into bubbles by their size and staff-to-resident ratio.** Grouping care homes into bubbles by their size and staff-to-resident ratio led to the same results as forming random bubbles.

## Impact of understaffing due to not using bank/agency staff

Filling vacant positions with bank/agency staff led to more infections (Fig 4) and outbreaks (Fig 5) than leaving these positions unfilled. When 10% of positions in care homes were unfilled, the average cumulative number of infections after 90 days were 33 per 1,000 residents (95%CI 33–34). When filling these vacant positions with bank/agency staff, the average cumulative number of infections increased to 40 infections per 1,000 residents (95%CI 39–40). When the vacant positions increased to 20% of total staff, leaving these positions unfilled resulted in an average of 35 infections per 1,000 residents after 90 days (95%CI 34–35). Filling

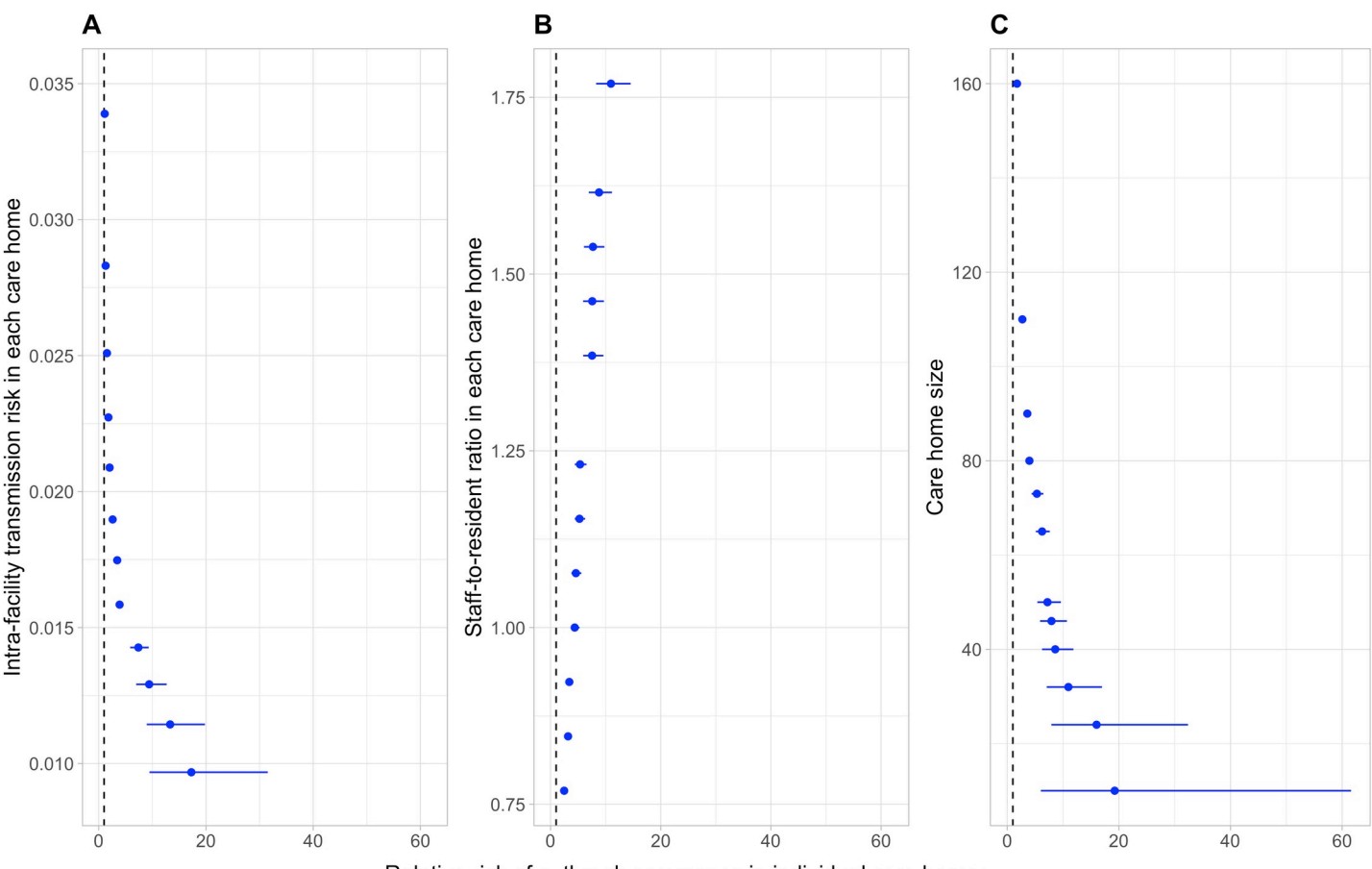

**Fig 3. Impact of using bank/agency staff upon individual care homes with different characteristics.** The plot illustrates the relative risk [RR] of outbreaks (points) within 90 days in individual care homes using 10% bank/agency staff compared with those care homes not using bank/agency staff. A: Care homes in network B (homogeneous size & staff-to-resident ratio) with heterogeneous transmission risk drawn from a Beta distribution (shape 1 = 5, shape 2 = 266). B: Care homes in network C (homogeneous size & heterogeneous staff-to-resident ratio) with homogeneous transmission risk. C: Care homes in network D (heterogeneous size & homogeneous staff-to-resident ratio) with homogeneous transmission risk. No intervention in bank/agency staff is implemented. Line range denotes the 95% CI of the RRs. Dashed blacked vertical line denotes the RR of 1.00.

these positions with bank/agency staff led to an average of 51 infections per 1,000 residents (95%CI 50–52). The stochasticity of bank/agency staff movement across care homes in the network led to wider distributions of infections as compared to leaving these positions unfilled (Fig 4). The stochasticity of bank/agency staff movement also led to a larger variation in outbreak probabilities (Fig 5B as compared to Fig 5A) as the proportion of vacant positions to total staff increased.

## Validation results

**Black-Box validation.** *Pattern i:* Our modelling results on the RR of infection for residents and staff in care homes that used bank/agency staff compared to ones that did not were consistent with other studies'. The Vivaldi study, conducted in more than 9,000 care homes in England, reported that the odds ratio [OR] of infection for staff in care homes using bank/agency staff compared with care homes not using these staff was 1.88 (95%CI 1.77–2.00) among care homes in London. Bank/agency staff constitute 20% of the adult social care workforce in the London region [6]. This study did not include information on the compliance rate

A: Care homes are understaffed

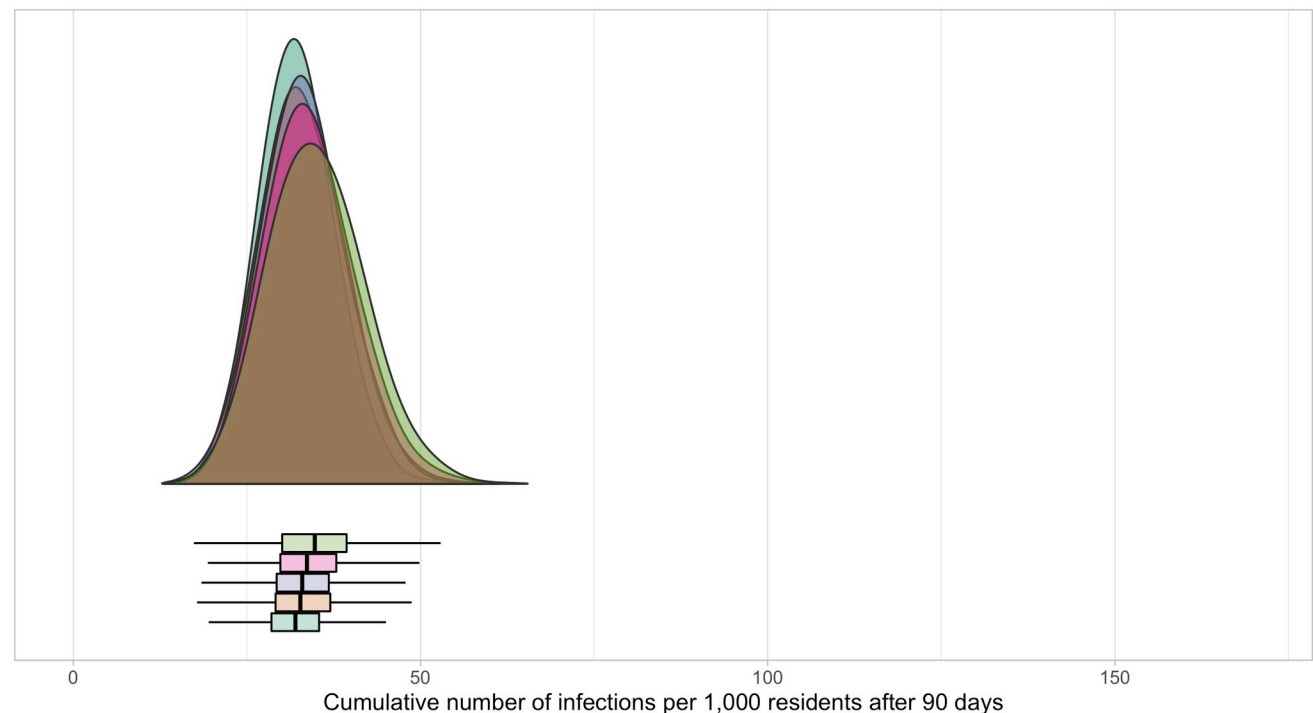

B: Care homes use bank/agency staff

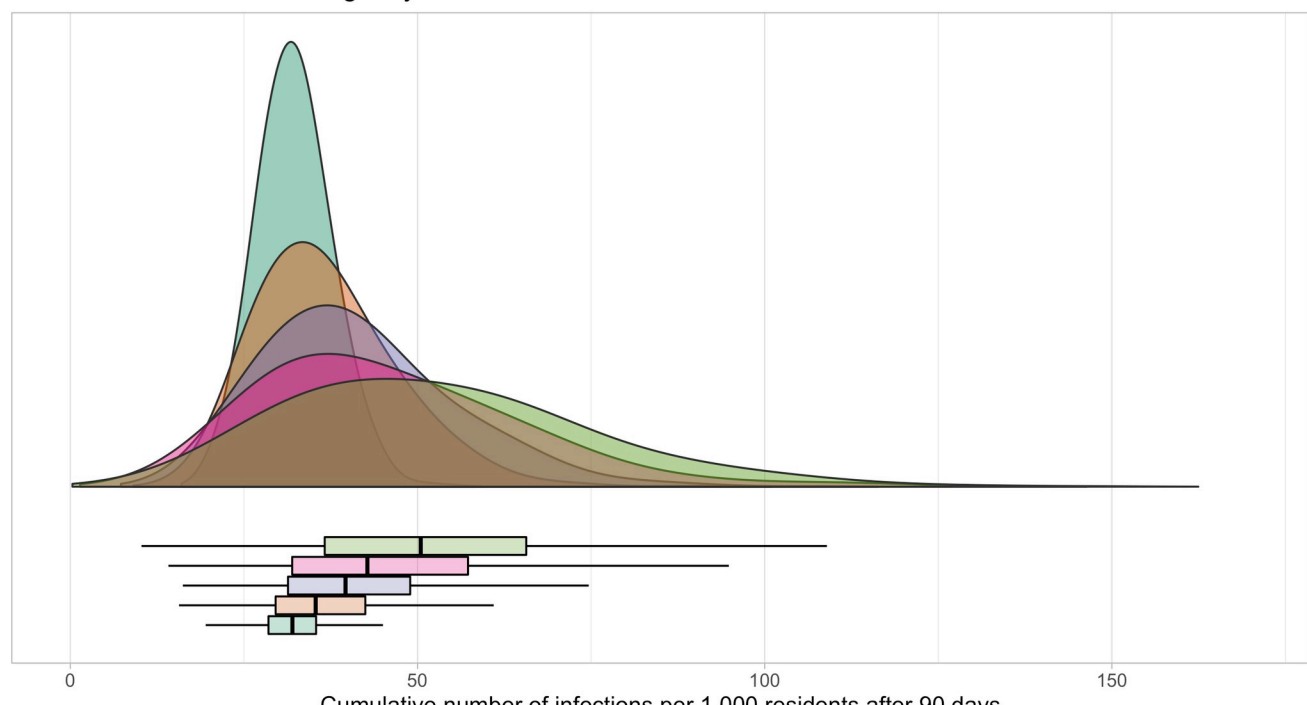

**Fig 4. Impact of staffing shortage vs use of bank/agency staff on infections in residents.** The plots illustrate the distributions of cumulative number of infected residents after 90 days. A–in various levels of staff shortage. No bank/agency staff are used to cover the vacant positions. B–in various usage levels of bank/agency staff. Bank/agency staff have 80% compliance to weekly PCR testing ($\alpha = 0.5$). Boxplot: middle–median; lower hinge– 25% quantile; upper hinge– 75% quantile; lower whisker = smallest observation greater than or equal to lower hinge—1.5 * IQR; upper whisker = largest observation less than or equal to upper hinge + 1.5 * IQR.

to weekly testing in bank/agency staff. The Health and Social Care Partnership (HSCP) Lanarkshire and Public Health Scotland advised that the compliance of bank/agency staff was likely to be lower than among permanent staff, which ranges between 70% and 90%. For the same level of bank/agency staff use, the corresponding OR in our model was very close to the Vivaldi study's finding (OR 1.81, 95%CI 1.77–1.86) for 60% testing compliance among bank/agency staff. Our model also approximated well the increased risk of infection for residents (Model: OR 2.20, 95%CI 2.14–2.27 (60% testing compliance); OR 1.67, 95%CI 1.65–1.70 (80% testing compliance); Vivaldi: OR 1.58, 95%CI 1.50–1.65) [2].

*Pattern ii*: Our model reproduced a similar increase in the risk of infection for staff working across multiple care homes compared to staff working in single care homes. A study

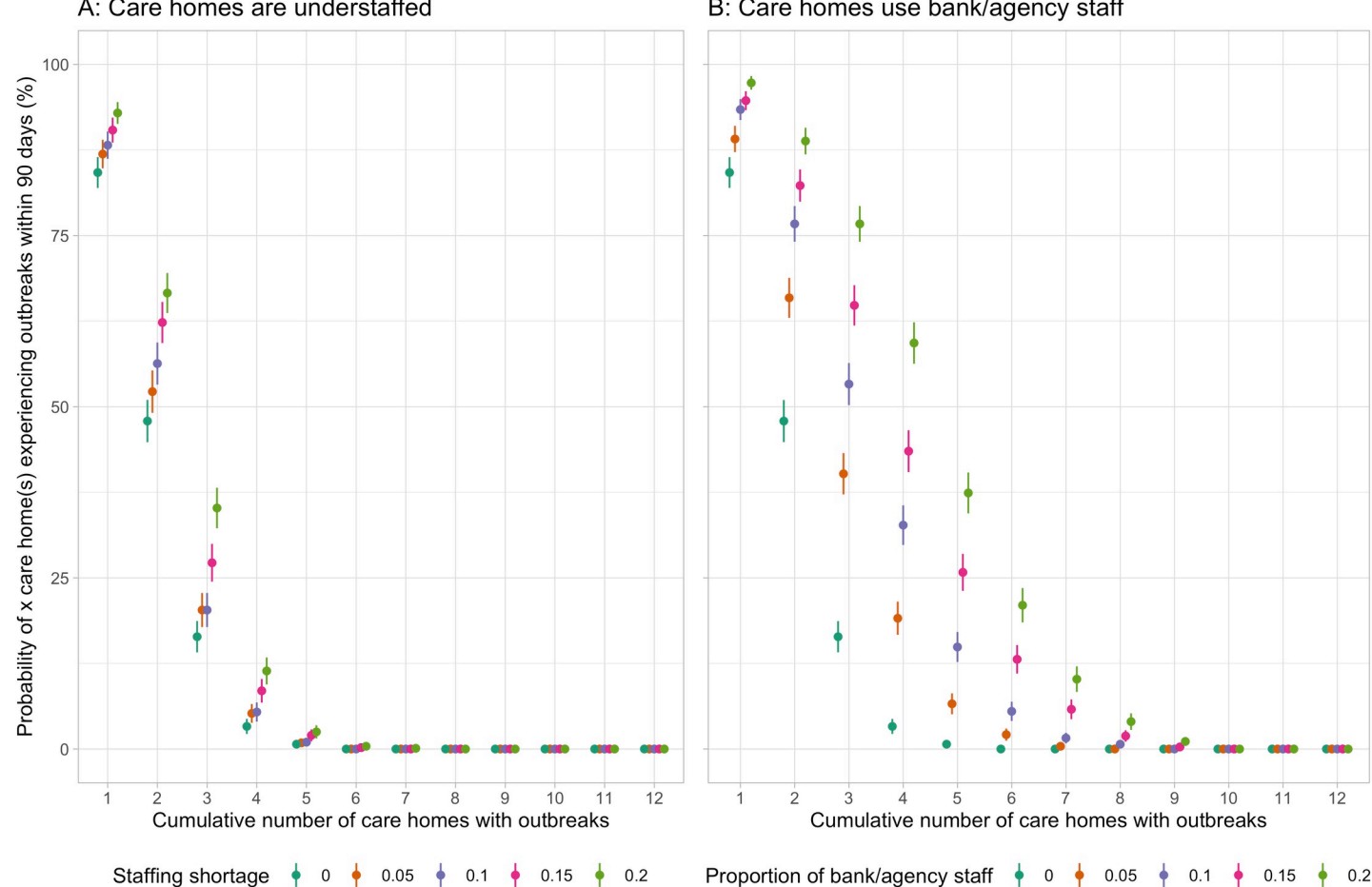

**Fig 5. Impact of staffing shortage verse use of bank/agency staff on the risk of outbreak.** The plots illustrate the risk of outbreak occurrence across care homes within 90 days. A–in various levels of staff shortage. No bank/agency staff is used to cover the vacant positions. B–in various usage levels of bank/agency staff. Bank/agency staff have 80% compliance to weekly PCR testing ($\alpha = 0.5$). The risk of outbreak occurrence (point) is the proportion of simulations where outbreaks occur in 1,000 simulations for each scenario. Line range denotes the 95% CI of this outcome.

conducted in April 2020 in six London care homes where 10.6% of staff had worked across multiple care homes reported that these staff had a three-fold higher risk of infection than staff working in single care homes (95%CI 1.90–4.79) [1]. In this period, no testing was conducted and the intra-facility transmission risk was likely higher due to PPE shortages and lack of clear and consistent IPC guidance. For a similar level of bank/agency staff use and when neither permanent nor bank/agency staff were tested weekly, the corresponding RR in our study was 3.09 (95%CI 3.03–3.15).

*Pattern iii*: Our model also reproduced a similar increase in the risk of outbreaks due to using bank/agency staff (Model: OR 3.42, 95%CI 3.25–3.60 (60% testing compliance); OR 2.53, 95%CI 2.40–2.66 (80% testing compliance); Vivaldi: OR 2.33, 95%CI 1.72–3.16) [2]. This finding was in line with another study in 34 Liverpool care homes. The study reported that care homes employing agency staff had an increased risk of Covid-19 outbreaks (RR 8.4, 95% CI 1.2–60.8) [38]. Due to these study results' wide confidence intervals, we only compared the trend qualitatively. This finding coincided with the results from Baister et al, who used a compartment model to simulate the spread of Covid-19 in the Lothian health board (Scotland) [39].

*Pattern iv*: The association between care home characteristics, including resident population size and staff-to-resident ratio, and Covid-19 outbreaks in our model echoed the results from other observational studies. Our model results showed that the care home population size was strongly associated with a Covid-19 outbreak (OR per 20-bed increase 2.32, 95%CI 2.15–2.51). This finding was consistent with the trend observed in the data of care homes in the UK and the finding from the investigation of 189 care homes in Lothian (OR per 20-bed increase 3.35, 95%CI 1.99–5.63) [40,41]. We found no association between the staff-to-resident ratio and the risk of outbreaks (Pearson's correlation coefficient –0.17, 95%CI –0.45–0.68) (S4 Appendix). This finding was in line with the finding in care homes in Liverpool [38].

**Sensitivity and uncertainty analyses.**  We summarize the outputs from the PRCC analyses in Table A in S4 Appendix. The per-contact transmission risk and the infection incidence in the community were the most significant contributors to the uncertainty in the number of infections in residents and staff. Increasing these parameters resulted in an increase in the number of infected residents and staff. These parameters were positively associated with the number of infected residents and staff. The number of infections was also sensitive to the staff-resident contact rate and the duration of pre-symptomatic disease but to a significantly lesser extent. The RR of infection for bank/agency staff to permanent staff was only sensitive to the per-contact transmission risk.

The relative effectiveness of interventions targeting bank/agency staff remained robust to modifying network composition in terms of care home size and staff-to-resident ratio, heterogeneity in intra-facility transmission risk, and the number of care homes in the network. The only exception was that forming bubbles of care homes was no longer effective in the scenario with both no weekly testing for bank/agency and different intra-facility transmission risks drawn from the Beta distributions (KS tests: p>0.1).

## Discussion

Our model provides a unique perspective on exploring the impacts of bank/agency staff movement on the spread of Covid-19 across care homes in a network. Combining ABM for the inter-facility transmission and SD for the intra-facility transmission allows us to effectively address research questions that are difficult to study with a single method. ABM enables the heterogeneity of care homes and the stochasticity of bank/agency staff's movement that characterize the inter-facility spread of Covid-19 to be captured. This bottom-up modelling

method is also more flexible than SD to reflect changes in the network composition, and it allows us to explicitly model interventions such as creating bubbles. Meanwhile, stochastic SD, with a lower computational intensity than ABM, provides a holistic perspective of the transmission dynamics in each care home which sufficiently meets our modelling objectives.

Consistent with other Covid-19 prevalence surveys in care homes in the UK [1,2,38], our findings generally support policies limiting the movement of staff working across multiple care homes if their testing compliance is low. Our modelling study found that the usage level of bank/agency staff in care homes significantly impacts the risk of SARS-CoV-2 infection in residents and the risk of outbreaks in care homes. Bank/agency staff working across multiple care homes can act as vectors that facilitate the inter-facility transmission of Covid-19. Additionally, the lack of infection control training and accountability among bank/agency staff, and their unfamiliarity with various practice protocols across care homes potentially limit their capability to adhere to the IPC procedures [44]. This undermines the implementation of IPC and increases the risk of infection for residents and staff.

We explored weekly testing of bank/agency staff in this paper and we found that it reduced the spread of Covid-19 across care homes. Increasing compliance to routine testing among these staff reduces the risk of infection among residents and the risk of outbreaks in the care homes. Our previous study showed that increasing the frequency of routine testing for staff within care homes is likely even more effective in reducing infections [24], though it may lead to reduced compliance. Despite the effectiveness of testing when compliance is high, residents in care homes using bank/agency staff are still exposed to a slightly higher risk of infection compared with those care homes not using bank/agency staff.

The effect of using bank/agency staff on the risk of an outbreak varies with a care home's relative–compared to other care homes in the network–intra-facility transmission risk, staff-to-resident ratio, and size. Bank/agency staff are more likely to acquire the infection in care homes with a higher transmission risk and then spread it into care homes with a lower transmission risk than in the reverse direction. Care homes with higher transmission risks are also more likely to experience an outbreak before exporting the virus via bank/agency staff. Therefore, the simulation results indicated that using bank/agency staff increased the risk of outbreaks in care homes with a lower transmission risk more significantly than in ones with a higher transmission risk. The risk of outbreaks increases significantly with the increase of care home size due to the increased risk of infection ingress by a larger number of staff members. Similar to the pattern observed in the network containing care homes with heterogeneous transmission risk, using bank/agency staff is more impactful on the risk of outbreaks in smaller care homes. Care homes with higher staff-to-resident ratios have a higher average daily number of bank/agency staff on duty, which increases the risk of infection ingress via this route and, thus, the risk of outbreaks.

Creating care home bubbles within which bank/agency staff work had a limited effect on simulation infection estimates. This intervention slightly reduces the number of infections when bank/agency staff's compliance to weekly testing is low. In other scenarios, forming bubbles adds no value to reduce the risk of infection for residents. Testing bank/agency staff weekly, to quickly identify asymptomatic and pre-symptomatic staff, prevents them from spreading the infection whether they are in bubbles or not. Whether care homes are grouped randomly or based on their characteristics such as size and staff-to-resident ratio does not affect the overall number of infections in the network. The latter approach shifts the risk of outbreaks and the number of infections, reducing them in smaller and higher staff-to-resident ratio care homes but increasing them further in larger and lower staff-to-resident ratio ones. However, this approach may cause a mismatch between the demand and supply of bank/agency staff. For example, bank/agency staff may resist working in bubbles where care homes

have a higher risk of outbreaks or are distant from each other. The practicalities of this approach need to be explored and discussed with HSCPs and care homes.

Our model estimates also suggest that staff shortages increase the risk of infection for residents and the risk of outbreaks across care homes to a lesser extent than using bank/agency staff to fill vacant positions. However, these estimates underestimate the impact of staff shortages as we did not account for their potential to reduce compliance to IPC measures leading to an increased transmission risk. Furthermore, staff shortages negatively impact the quality of care delivered to residents and undermine the effort to resume visitation, which affects residents' well-being [45–47]. Staff shortages also increase the workload and pressure on staff, potentially causing them mental and psychological strain and can make them leave their positions. Staff burnout and its impact on controlling outbreaks and reducing compliance with IPC practices should be researched further.

Despite our findings, employing bank/agency is viable to care homes to avoid falling short of safe staffing levels and losing places for residents. The link between the use of bank/agency staff and poorer care outcomes and higher risks of infection have been reported in previous studies, and the latter is echoed by our modelling study. However, as other research has noted, care homes could potentially mitigate these risks by increasing wages, offering incentives for working in single care homes, and offering sick leave [4,48]. As our model suggests that routine testing of bank/agency staff significantly reduces the risk of Covid-19, care homes utilising bank/agency staff and agencies supporting care homes during the pandemic may wish to consider protocols and support to enhance compliance to the testing intervention in this group of workers. In the longer term, better pay and training, including IPC training, will help create a higher quality and more stable workforce.

Our modelling study has a number of limitations. Firstly, we assumed that when bank/agency staff are in care homes, they have the same compliance level to IPC measures as permanent staff, including hand hygiene, wearing PPE, and social distancing. If bank/agency staff compliance is lower, our study underestimates the increased risk of Covid-19 transmission by them. Secondly, the model has not accounted for the activities that bank/agency staff undertake within the care homes. These activities would affect their contact rates and the nature of their contacts, which in turn influences the per-contact transmission risk with residents. The evidence remains uncertain on the difference in the risk of SARS-CoV-2 infection between care home staff working in resident-facing roles compared with those not working in these roles. The estimates in a Covid-19 infection survey in England showed evidence of an increased level of infection amongst staff working in resident-facing roles [49]. However, other studies suggested infection rates among staff members within individual care homes do not statically differ when comparing different exposures to the residents, including those with no contact with residents [1,50]. Thirdly, our model has not accounted for the potential increased risk of SARS-CoV-2 infection in bank/agency staff who also work in other healthcare settings such as hospitals and/or carry out other care duties. Interactions in these settings typically require closer contact than in the community more broadly. Fourthly, the model has not considered scenarios in which bank/agency staff move across care homes multiple times per day. These factors would serve to increase the impact of using bank/agency staff. Finally, we have not modelled care homes' adaptive decisions about interventions which can also contribute to affect the intra-facility transmission risk. For example, care homes that experience outbreaks may become more compliant to IPC measures while the compliance in other care homes that have not had outbreaks may decrease over time.

Several key parameters describing virus and disease characteristics are still uncertain and may also vary greatly from community to community. Similarly, a lot of relevant information about the characteristics of care home resident population and staff are not readily available.

Therefore, we performed sensitivity and uncertainty analyses for a wide range of parameter values and various model characteristics respectively. The purpose of this modelling study was not to project the absolute number of SARS-CoV-2 infections and deaths in residents and staff in care homes, rather to compare the relative effectiveness of different interventions targeting staff working across multiple care homes. Although the absolute values of the model outcomes are sensitive to some parameters and changes in model structure, the relative findings which have been our focus are robust to uncertainty in model parameters and structure. Our results also help understand how heterogeneity in network composition affects individual care homes. Our base case model reflected characteristics of care homes in the UK but it could be tailored to a specific network of care homes in other countries to evaluate the impacts of policies targeting staff working across multiple care homes. The model can also be updated and extended to reflect the heterogeneity in care homes' adaptive decisions about interventions.

## Conclusion

In conclusion, this modelling study has implications for policymakers considering developing effective interventions targeting staff working across multiple care homes during the ongoing and for potential future pandemics. The use of bank/agency staff working in multiple care homes increases the risk of SARS-CoV-2 infection for residents and the risk of outbreaks across these facilities. Our results suggest that the movement of staff across care homes should be limited and care homes should use bank/agency at a minimum possible level to reduce infections. Where using bank/agency staff is unavoidable, they must be encouraged to comply with routine testing. They should also be inducted into new care home environments to enhance their compliance to other IPC interventions. Forming bubbles of care homes shows little value in reducing the risk of inter-facility transmission and may be resource-consuming to implement and monitor.

## Supporting information

**S1 Appendix. The ODD Protocol. Table A. The state variables of care homes agents and bank/agency staff agents**. **Fig A. The process overview and scheduling of the model at each time step**. (White box: sub-model; sky-blue box: starting at time step 1; green box: starting at time step 91) Note: FoI–Force of infection. **Fig B. The progression of COVID-19 infections**. Susceptible people may acquire the infection when exposed to infectious sources. They are infected but not yet infectious (exposed state). Once exposed people become infectious, they can either remain asymptomatic for the entire infectious period or develop symptoms after a pre-symptomatic period. Symptoms could be mild or severe and require hospitalizations. Infectious people will eventually recover or die. **Table B. Resident population size and staffing level in care homes within a network**. **Table C. Initial values of entities' state variables and parameters**. **Table D. Parameters used in the model**. **Fig C. The structure of Intra-facility module embedded in each care home agent**. This sub-model developed using stochastic system dynamics represents the transmission dynamics of COVID-19 within a care home. Dash red, blue, and green lines represent transmissions caused by infectious permanent staff, residents, and temporary bank/agency staff respectively. **Table E.** Summary of equations for stocks and flows in the Intra-facility sub-model.
(DOCX)

**S2 Appendix. Methods. Fig A. Comparisons of results generated from parallel system dynamics [SD], stochastic SD, and agent-based [ABM] models**. The figure describes the time series of Covid-19 prevalence among residents in care home with capacity of 80 residents

(1,000 simulations per scenario). Base-case parameters are used. Interventions implemented in the care home include testing upon admission of residents, no visitation, hand hygiene and using PPE, social distancing, isolation of symptomatic/confirmed residents, and weekly testing of staff. Simulations are seeded with one infected resident. Box-plot: lower hinge: 25% quantile; lower whisker: smallest observation greater than or equal to lower hinge − 1.5×IQR; middle: median; upper hinge: 75% quantile; upper whisker: largest observation less than or equal to upper hinge + 1.5×IQR). Note: IQR–interquartile range. **Fig B. Time series of Covid-19 prevalence among residents**. (A) and cumulative number of infected residents after 90 days (B) with different values of intra-facility transmission risk. The figure describes the model outcomes for network B in three scenarios: The intra-facility per-contact transmission risk is i/ "Const": homogeneous across care homes (0.02); ii/ heterogeneous across care homes and drawn from Beta distribution (5, 266); iii/ heterogeneous and drawn from Beta distribution (2, 117). No intervention in bank/agency staff is implemented. Bank/agency staff comprises 10% of total staff. Other parameters have the base-case values. Boxplot: middle–median; lower hinge– 25% quantile; upper hinge– 75% quantile; lower whisker = smallest observation greater than or equal to lower hinge—1.5 * IQR; upper whisker = largest observation less than or equal to upper hinge + 1.5 * IQR. **Table A. Relevant studies for black-box validation identified from a systematic search**
(DOCX)

**S3 Appendix. Additional Modelling Results. Table A. Risk of infection in residents and staff in various usage levels of bank/agency staff**. **Fig A. Relative risk [RR] of infection for bank/agency staff with different compliance rates to weekly PCR testing to permanent staff in care homes using bank/agency staff**. Compliance to weekly testing among permanent staff is 80%. Results are for 1,000 simulations in each scenario. Boxplot: middle–median; lower hinge– 25% quantile; upper hinge– 75% quantile; lower whisker = smallest observation greater than or equal to lower hinge—1.5 * IQR; upper whisker = largest observation less than or equal to upper hinge + 1.5 * IQR.
(DOCX)

**S4 Appendix. Results of Sensitivity and Uncertainty Analyses. Table A. Output from Partial Rank Correlation Coefficient analyses**. **Fig A. Impact of staff-to-resident ratio and resident population size on risk of outbreak**. The plot describes the risk of outbreak occurrence within 90 days in individual care homes with A: the same population size of 65 residents but different staff-to-resident ratios (network C). B: different resident population size (network D). The average intra-facility transmission risk in care homes is homogeneous. The average usage level of bank/agency staff is 10% of total staff. No intervention on bank/agency staff is implemented. The risk of outbreak occurrence (point) is the probability of simulations where outbreaks occur in 1,000 simulation for each scenario. Line range denotes the 95% CI of this outcome.
(DOCX)

## Acknowledgments

We thank Health and Social Care Lanarkshire (Trudi Marshall and Dennis McLafferty), and care homes in Lanarkshire for facilitating data collection. We also thank Professor Graham Ellis from the Chief Medical Officer in Scotland for Ageing and Health and the Data, Analysis & Research group for helpful discussion.

## Author Contributions

**Conceptualization:** Le Khanh Ngan Nguyen, Itamar Megiddo, Susan Howick.

**Data curation:** Le Khanh Ngan Nguyen.

**Formal analysis:** Le Khanh Ngan Nguyen.

**Funding acquisition:** Itamar Megiddo.

**Investigation:** Le Khanh Ngan Nguyen.

**Methodology:** Le Khanh Ngan Nguyen, Itamar Megiddo, Susan Howick.

**Project administration:** Le Khanh Ngan Nguyen, Itamar Megiddo.

**Software:** Le Khanh Ngan Nguyen.

**Supervision:** Itamar Megiddo, Susan Howick.

**Validation:** Le Khanh Ngan Nguyen, Itamar Megiddo, Susan Howick.

**Visualization:** Le Khanh Ngan Nguyen.

**Writing – original draft:** Le Khanh Ngan Nguyen.

**Writing – review & editing:** Itamar Megiddo, Susan Howick.

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
