## [Decision Letter · Decision Letter 0]

8 Nov 2021

Dear Ms Nguyen,

Thank you very much for submitting your manuscript "Hybrid simulation modelling networks of heterogeneous care homes and the inter-facility spread of covid-19 by sharing staff" for consideration at PLOS Computational Biology.

As with all papers reviewed by the journal, your manuscript was reviewed by members of the editorial board and by several independent reviewers. In light of the reviews (below this email), we would like to invite the resubmission of a significantly-revised version that takes into account the reviewers' comments.

We cannot make any decision about publication until we have seen the revised manuscript and your response to the reviewers' comments. Your revised manuscript is also likely to be sent to reviewers for further evaluation.

Sincerely,

Claudio José Struchiner, M.D., Sc.D.

Associate Editor

PLOS Computational Biology

Virginia Pitzer

Deputy Editor-in-Chief

PLOS Computational Biology

Reviewer's Responses to Questions

**Comments to the Authors:**

Reviewer #1: This paper develops a model of the spread of covid infections in care homes and it explores a number of relevant scenarios related to staffing, with particular interest to the use of bank staff. This is a topical issue with the potential to inform decisions in staffing of care homes and it is worthy of publication. The paper is well designed and written and the models well implemented. Some suggestions for further improvement are listed below:

1. The paper discusses the use of hybrid simulation in the introduction. At this point the justification of using a hybrid model does not work very well. It refers to the use of hybrid modelling, but it is not clear for the reader why hybrid simulation is suitable. It would be better if the authors explain in the introduction the problem and the key aspect that the model needs to represent. The justification of the use of hybrid modelling could be discussed in the materials and methods section.

2. It seems like the problem is the lack of staff in care homes, which is only indirectly referred to in the paper. It would be useful to make this clearer from the beginning and in the abstract also.

3. What is the aim of the model? It would be useful to include in the methods section the aim and questions that the model aims to answer.

4. Results section - Impact of different usage levels of bank/agency staff. It would be useful to indicate whether there is a statistically significant difference in the RR of infection or outbreak between the different scenarios tested.

5. There are two sections on validation - Confidence building (Verification and validation) and Validation results. Consider merging.

6. There are some typos in the manuscript that need attention. For example line 80 – understaffed instead of understaffing. Please check the manuscript carefully.

Overall, i think this is very interesting work and worthy of publications. I look forward to seeing the revised manuscript.

Reviewer #2: This is a rather comprehensive study with complex modelling consideration on the disease transmission in a heterogeneously mixed system. I have the following concerns. The major one is that this manuscript seems to lack a real-world dataset to support the modelling outcomes.

. in table S1-5, the F*xi term in the transmission path seems strange to me. what is the meaning of (1 - F*xi)? And what is the intuition behind this term? Besides, it seems the physical unit does not hold consistently. The unit of all ‘flow’ should be ‘# of individuals’, right?

. should it be normal (0,1)? Normal(1,0) means just constant 1.

. is there any fitting conducted in this study? If yes, how is the fitting conducted? So far, I did not see the fitting results.

. please discuss the limitations of this study.

Reviewer #3: Please see the attachment.

Reviewer #4: See attached.

**Have the authors made all data and (if applicable) computational code underlying the findings in their manuscript fully available?**

Reviewer #1: Yes

Reviewer #2: **No: **I did not find code or data open-accessed.

Reviewer #3: Yes

Reviewer #4: **No: **Most of the simulations are conducted in a packed software AnyLogic so it's hard to verify the results, though the authors share the setting of the model.

PLOS authors have the option to publish the peer review history of their article (what does this mean?). If published, this will include your full peer review and any attached files.

Reviewer #1: No

Reviewer #2: No

Reviewer #3: No

Reviewer #4: No
---

## [Decision Letter · Decision Letter 1]

20 Dec 2021

Dear Ms Nguyen,

We are pleased to inform you that your manuscript 'Hybrid simulation modelling networks of heterogeneous care homes and the inter-facility spread of covid-19 by sharing staff' has been provisionally accepted for publication in PLOS Computational Biology.

Best regards,

Claudio José Struchiner, M.D., Sc.D.

Associate Editor

PLOS Computational Biology

Virginia Pitzer

Deputy Editor-in-Chief

PLOS Computational Biology

Reviewer's Responses to Questions

**Comments to the Authors:**

Reviewer #4: The authors addressed all my concerns well, and I recommend the manuscript for publication.

**Have the authors made all data and (if applicable) computational code underlying the findings in their manuscript fully available?**

Reviewer #4: **No: **

PLOS authors have the option to publish the peer review history of their article (what does this mean?). If published, this will include your full peer review and any attached files.

Reviewer #4: No

---

## [Editor Report · Acceptance letter]

7 Jan 2022

PCOMPBIOL-D-21-01703R1 

Hybrid simulation modelling of networks of heterogeneous care homes and the inter-facility spread of Covid-19 by sharing staff

Dear Dr Nguyen,

I am pleased to inform you that your manuscript has been formally accepted for publication in PLOS Computational Biology. Your manuscript is now with our production department and you will be notified of the publication date in due course.

With kind regards,

Katalin Szabo
